# Eye Gaze Tells You Where to Compute: Gaze-Driven Efficient VLMs

## Abstract

Vision-Language Models (VLMs) deliver impressive performance in understanding visual content with language instructions. However, redundancy in vision tokens results in the degenerated inference efficiency of VLMs, which hinders real-time use on edge consumer devices such as Virtual Reality (VR) headsets and Augmented Reality (AR) glasses. Existing efficiency methods commonly prune visual tokens using learned saliency, sparse attention schedules, or controller policies, but they often require architectural modification or access to intermediate activations. These pipelines add inference-time modules that increase compute and memory and often lead to an accuracy trade-off. Moreover, they also suffer from misalignment between the prompts and the region of interest in the images. Without human guidance, the model may focus on the wrong regions and miss small, high-frequency details when prompts or scenes change. In this paper, we propose GazeVLM, a training-free framework that uses the human eye gaze as a natural supervisory signal to allocate computation where it matters. By extracting gaze-driven regions of interest (ROIs) and optionally combining them with a low-resolution global view, GazeVLM mimics fovea-periphery perception to cut redundant visual tokens while preserving task-relevant details. We evaluate the visual question answering tasks on Qwen2.5-VL-3B/7B on the VOILA-COCO benchmark with human gaze. Quality of the answer is assessed by GPT-4o pairwise judging and a weighted score over coverage, accuracy, details, and fluency. Efficiency is measured by token counts and FLOPs. GazeVLM reduces visual tokens by up to 93.1%, total tokens by up to 59.6%, and FLOPs by 50%, while keeping better answer quality relative to full-resolution baselines. Our results show that aligning model computation with human gaze offers a simple, plug-and-play path toward efficient VLM inference on consumer devices.

## 1 Introduction

Vision-Language Models (VLMs) demonstrate remarkable capabilities in tasks from basic image captioning (Bai et al., 2025) to advanced visual question answering (Hartsock & Rasool, 2024), and video comprehension (Weng et al., 2024). These advances have led to the integration of VLMs into consumer-facing devices where real-time multimodal understanding is crucial. In particular, augmented and virtual reality (AR/VR) platforms are increasingly incorporating VLMs to support natural and immersive interactions. For example, Meta has equipped the Orion AR glasses with VLM-powered assistants (Meta, 2024).

However, deploying VLMs on such resource-constrained AR/VR devices faces several challenges. First, visual input tokenization introduces substantial computational overhead. Unlike text tokens, which are compact, visual information is sparsely distributed within dense pixel arrays. As a result, visual encoders must process large numbers of patches, most of which correspond to semantically irrelevant background. OpenAI's analysis (OpenAI Community, 2023) reports that a single 1024×1024 image requires 765 tokens, which is over an order of magnitude more than typical text queries. Therefore, vision-related computation can account for more than 90% of the total cost in image captioning. Second, this overhead translates directly into high energy consumption, which is a critical constraint for battery-powered edge devices. Meta reports that a 7B-parameter VLM model consumes about 0.7 J per token (Liu et al., 2024), meaning that processing a single 1024×1024 image (765 tokens) requires over 500 J of energy. The battery capacity of consumer AR glasses varies

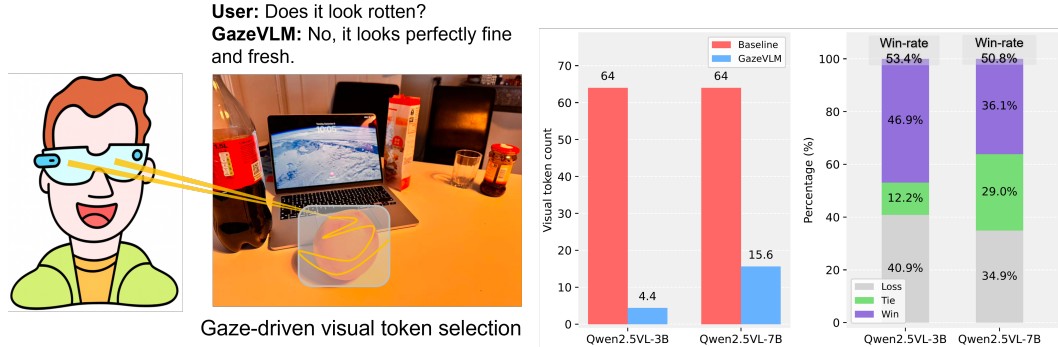

Figure 1: GazeVLM at a glance. GazeVLM uses the user's eye gaze to form an adaptive foveated input, so the base VLM spends tokens where the user looks. On VOILA-COCO with Qwen2.5-VL-3B/7B, GazeVLM cuts visual tokens from 64 to 4.4 by 93.1% and 64 to 15.6 by 75.6%, respectively, while achieving pairwise win rates of 53.4% and 50.8% against full-image inference (ties excluded).

significantly, but typically provides less than 10 kJ of energy, whereas the VR headsets' battery size is large, for example, Apple Vision Pro (Apple Inc., 2024), which employs an external battery pack exceeding 100 kJ. This implies that most AR/VR devices can sustain inference for only a few frames, and even the Vision Pro can only handle around two hundred images before its battery is exhausted.

Interestingly, most of the recent AR/VR devices (Varjo, 2025; Apple Inc., 2024; Pico Interactive, 2024) are already equipped with, or are moving toward integrating, eye-tracking modules (Lin et al., 2025; Ding et al., 2025; Tan et al., 2025; Chen et al., 2023), which are primarily used for intuitive interaction and gaining insights into users' attention. This provides a unique opportunity to exploit human physiological signals for efficient inference of VLMs. Human gaze is a direct behavioral manifestation of visual attention, reflecting perceptual and cognitive processes that determine which regions of a scene are most relevant for the task. Unlike artificial saliency models (Han et al., 2025), gaze encodes real-time, user-specific intent and thus offers a natural supervisory signal for multimodal inference.

In this work, we study how human eye gaze signals can guide the efficiency of VLMs. We propose GazeVLM, a training-free framework that leverages the eye gaze to crop task-relevant regions of interest (ROIs), thereby reducing redundant vision tokens while preserving sufficient semantic coverage. Our approach is plug-and-play, requires no retraining, and aligns human and model attention to achieve substantial efficiency gains in VLM inference with even better accuracy in visual question answering tasks. To the best of our knowledge, our approach is the first work that leverages human eye gaze to guide visual token reduction. As shown in Fig.1, GazeVLM cuts visual tokens from 64 to 4.4 by 93.1% and 64 to 15.6 by 75.6%, respectively, on VOILA-COCO (Yan et al., 2024) with Qwen2.5-VL-3B/7B, while achieving pairwise win rates of 53.4% and 50.8% against inference with full-resolution input (ties excluded).

## 2 RELATED WORKS

VLMs, such as LLaVA (Liu et al., 2023), InstructBLIP (Dai et al., 2023), and Qwen-VL (Bai et al., 2025), have become a cornerstone for multimodal understanding by integrating vision encoders with pre-trained language models using projection layers or adapters, and are then fine-tuned on various vision-language tasks. These models achieve strong performance by representing images as sequences of visual tokens, but their inference cost grows quadratically with the number of tokens, underscoring the pressing need for more efficient processing strategies.

Many vision token selection or compression methods have been proposed recently. Sparse-VLM (Zhang et al., 2024a) considers text tokens from language instruction to guide the pruning of vision tokens. MMTok (Dong et al., 2025) formulates token selection as a maximum coverage problem and leverages multimodal similarity to select informative subsets. Event-priori VLM (Qin

et al., 2025) incorporates event-based priors to guide token selection for efficient visual understanding. OmniVLM (Chen et al., 2024) compresses visual tokens into compact representations using a multi-layer-perception (MLP) projector. However, they inevitably incur performance degradation when compressing and pruning vision tokens and typically operate solely on either vision or text cues.

A growing number of research explores incorporating human physiological signals (Yan et al., 2024; Wang et al., 2024; Lopez-Cardona et al., 2025; Zhang et al., 2024b; Lee et al., 2025), such as eye gaze and neural activity, into LLM systems. A recent survey (Lopez-Cardona et al., 2025) emphasizes that eye movements, EEG, and other cognitive signals provide direct behavioral manifestations of attention, reflecting perceptual and cognitive processes. However, these approaches primarily emphasize alignment and user experience rather than computational efficiency. In contrast, our work exploits gaze as a direct signal to guide efficient inference. We propose GazeVLM, a training-free framework that uses the gaze to crop ROIs, thereby reducing redundant vision tokens while preserving semantic coverage. This distinguishes our approach as both user-centered and efficiency-oriented, offering plug-and-play performance improvements without retraining. To the best of our knowledge, we are the first to introduce eye-gaze as a natural supervisory signal to allocate computation in VLMs.

## 3 INSIGHTS

Human visual attention is inherently sparse: at any moment, the eyes fixate on only a small fraction of the visual field, while the brain integrates peripheral information to form a coherent percept. Eye gaze thus serves as an external, high-fidelity signal of where humans allocate perceptual and cognitive resources. A rich body of cognitive science has established that gaze is tightly coupled with both perception and reasoning processes (Yarbus, 1967; Hoffman & Subramaniam, 1995). People tend to look at objects they are reasoning about, and fixations often precede verbal descriptions or answers to questions (Just & Carpenter, 1980; König et al., 2016).

This suggests that gaze is not arbitrary but reflects a user's semantic priorities in a scene. For example, when asked a visual question, human observers naturally direct their gaze toward the relevant region before responding (Tanenhaus et al., 1995). In contrast, existing Vision-Language Models (VLMs) distribute computation uniformly over dense image patches, processing both foreground and background with equal effort. This uniform treatment is misaligned with human perception: semantically irrelevant regions consume the same computational budget as task-relevant foreground. Shown in Fig. 2, task-relevant foreground is not fixed, it depends on the current reasoning goal. Human gaze dynamically shifts toward different regions of the scene, meaning that what counts as foreground in one context may become background in another. Thus, unlike static saliency, gaze reflects task-dependent priorities.

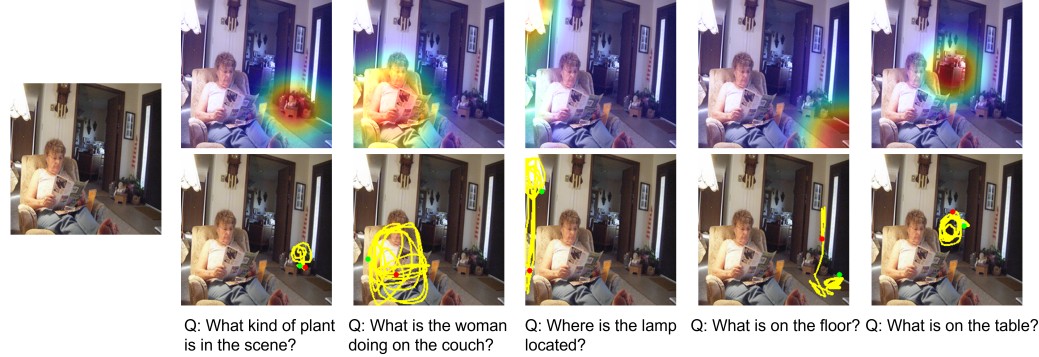

Q: What kind of plant is in the scene?   Q: What is the woman doing on the couch?   Q: Where is the lamp located?   Q: What is on the floor?   Q: What is on the table?

Figure 2: Human perception of a scene is task-dependent: what serves as foreground in one context may become background in another. Eye gaze trace and corresponding gaze heatmap (Yan et al., 2024) show the attention shifts.

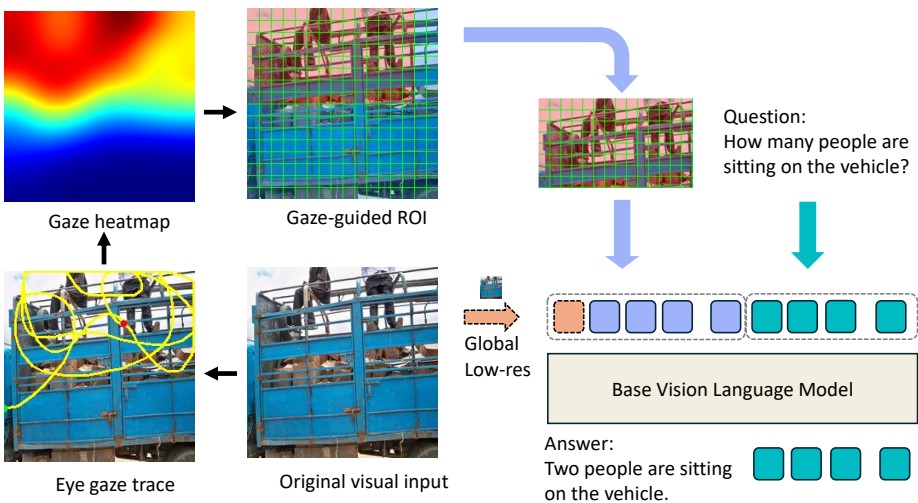

Figure 3: Overview of our proposed GazeVLM. Given an input image and eye gaze traces, we compute a gaze heatmap and extract a compact ROI that preserves attended areas. The ROI is optionally combined with a downsampled global view and encoded into image tokens, which are processed by a base model. This design emulates human fovea-periphery perception, allocating higher resolution to attended regions while maintaining global context.

Our key insight is that by aligning model processing with human gaze, we can reduce redundant vision tokens while preserving the most informative content for multimodal reasoning.

## 4 METHOD

Based on the aforementioned insights, we propose GazeVLM, a training-free token-compressed inference method.

### 4.1 GAZE-DRIVEN ROI EXTRACTION

In order to efficiently compress the visual input, we first need to determine which region of the image is most relevant to the user's attention and the task at hand. Human gaze provides a natural signal for this purpose: when a subject looks at an image, their eye gaze trace indicates which parts of the scene they consider important. We leverage this signal to identify an ROI that should be processed at least at a higher priority.

Formally, let the input image be
$$I \in \mathbb{R}^{H \times W \times 3},$$
and its corresponding gaze heatmap
$$G \in [0,1]^{H \times W}, \quad \sum_{x,y} G(x,y) = 1,$$
where each value $G(x,y)$ indicates the likelihood that pixel $(x,y)$ is attended by the user. Intuitively, higher values in $G$ correspond to gaze-concentrated regions.

The raw traces are first mapped onto the image plane as a sequence of discrete points. To transform these sparse and noisy points into a continuous representation, a Gaussian smoothing is applied, spreading each point's influence over a local region. This produces a smooth heatmap that reflects the distribution of visual attention across the image.

To extract a compact ROI, we accumulate the most attended pixels until a fixed proportion $\rho$ of the total gaze mass is covered. Specifically, we sort all pixels in descending order of $G(x,y)$ and

identify the smallest support set $\mathcal{S}_\rho$ such that

$$\sum_{(x,y)\in\mathcal{S}_\rho} G(x,y) \geq \rho. \tag{1}$$

The minimal bounding box enclosing $\mathcal{S}_\rho$ defines the ROI:

$$B = (x_0, y_0, x_1, y_1).$$

In practice, directly cropping $B$ may lead to unstable results (e.g., too tight crops that miss context or degenerate cases when gaze is sparse). Therefore, we enforce a minimum crop size $R_{\min}$ to guarantee sufficient resolution.

This procedure yields a gaze-driven crop $I_r = I[B]$ that preserves the most relevant details while discarding irrelevant regions. Our method is directly grounded in human gaze, which naturally aligns with the semantic intent of the task.

## 4.2 Two-scale Input Representation

Humans do not process the entire visual field at uniform resolution. Instead, the fovea provides high-resolution vision at the attended region, while the periphery captures the broader scene at lower resolution.

Once the ROI is extracted, the next question is: *should we only feed the ROI to the model*? While the ROI contains the most task-relevant details, discarding the full image entirely would remove global cues such as scene layout, object co-occurrence, or background context, which may still be essential for correct reasoning. On the other hand, if we only feed the full image at a fixed resolution, the ROI can only be represented at low fidelity, making it difficult for the model to capture fine-grained details (e.g., small objects or text).

To resolve this trade-off, we introduce a Two-scale input strategy inspired by the fovea-periphery organization of the human visual system. Specifically, we construct two complementary image views:

- Global view ($I_g$): the original image is downsampled to a coarse resolution $H_g \times W_g$. This preserves the holistic scene layout at a low computational cost, analogous to peripheral vision.
- ROI view ($I_r$): the gaze-driven crop with resolution $H_r \times W_r$. This provides fine-grained detail in the attended region, analogous to foveal vision.

Both $I_g$ and $I_r$ are processed by the encoder to generate visual tokens, which are then concatenated with the text query $T$ to form the final model input:

$$X = \mathcal{P}\big(\{I_g, I_r\}, T\big), \tag{2}$$

where $\mathcal{P}$ denotes the process that aligns visual and textual modalities.

In practice, we guide the model by explicitly indicating that the second image corresponds to the ROI and should be prioritized during reasoning. This allows the model to combine global context from $I_g$ with high-resolution evidence from $I_r$.

Overall, the Two-scale representation mirrors human perception: the global view maintains peripheral awareness, while the ROI view supplies foveated detail. This design enables the model to allocate its limited token budget more effectively, improving both efficiency (fewer irrelevant tokens) and accuracy (sharper focus on gaze-attended content).

## 5 Experiments

**Experimental Setup.** To demonstrate the effectiveness of GazeVLM, we conducted experiments on Qwen2.5-VL-3B-Instruct and Qwen2.5-VL-7B-Instruct models, using the dataset VOILA-COCO (Yan et al., 2024). All experiments were conducted on one NVIDIA A100 GPU. For each sample, we smooth the eye-gaze trace into a heatmap, take the smallest region whose cumulative mass reaches $\rho$. We then build a Two-scale input by concatenating a downsampled global view ($H_g = 28, W_g = 28$) with the cropped ROI; the prompt explicitly states that the second image is the ROI, so the model prioritizes it during reasoning.

**Datasets.** VOILA-COCO is a gaze-aligned VQA dataset built from the Localized Narratives (LN-COCO) corpus using an automatic data annotation pipeline. It includes images, gaze traces, image captions, corresponding questions, and standard answers. The released splits contain 20k images with 70k QA pairs, respectively.

**Evaluation Metric.** As VOILA-COCO, we also employ GPT-4o as an automatic judge to evaluate the quality of model-generated answers. We adopt a pairwise comparison setting: given the question, the image caption, and the ground-truth answer, GPT-4o is asked to compare two candidate responses. To mitigate the sequence-order bias, we implement a dual-setting evaluation in which the order of the two answers is reversed, and the results are aggregated. Beyond a preference label, we also instruct GPT-4o to provide fine-grained scores on four dimensions: coverage (0-10), accuracy (0-10), details (0-10), and fluency (0-10). A weighted total score is computed as

$$\text{total} = 0.40 \times \text{coverage} + 0.40 \times \text{accuracy} + 0.15 \times \text{details} + 0.05 \times \text{fluency}.$$

This total score (0-10) reflects both alignment with the ground truth and language quality. We report GPT-4o scores, and also compute win-rates. Besides performance evaluation, we also quantify the computational cost by measuring the number of FLOPs to assess the efficiency.

### 5.1 Effect of ROI Selection on Efficiency and Accuracy

We study how much gaze mass to keep when forming the ROI, controlled by the threshold $\rho$. Intuitively, a small $\rho$ keeps only the highest-confidence gaze hotspots (tighter crops, extreme compression), while a large $\rho$ expands the crop to include more peripheral content (looser crops, more tokens).

As shown in Tab. 1, on Qwen2.5-VL-3B, even very aggressive pruning already helps. At $\rho = 0.05$, we reduce the visual tokens from 64 to 4.4 on average by 93% and reduce total FLOPs by roughly 50% from 267.6 GFlops to 132.8 GFlops, yet we still beat the full-image baseline with 53.4% win-rate. As we relax the crop, quality improves steadily, peaking near $\rho = 0.5$, achieving 68.2% win-rate over baseline, with visual tokens still reducing from 64 to 23.4. Once the crop becomes too large, for example, 27.8 visual tokens when $\rho = 0.6$, we start to reintroduce clutter, and we dilute pixel density on the truly relevant parts. In other words, beyond a point, adding more peripheral pixels trades foveal detail for background that the model did not need or the question did not need.

The 7B model follows the same curve but is shifted to the right. With very small $\rho$, 7B underperforms the full-image baseline, for example, 34.1% win-rate at $\rho = 0.05$. It crosses the 50% line only when the crop becomes moderately large (50.8% at $\rho = 0.3$ with visual token reducing to 15.6), and then plateaus around 51-53%. This asymmetry between 3B and 7B is obvious. The stronger baseline of the 7B model already handles global context and distractors better, over-tight crops remove relational cues like layout, counts, left/right that the larger model actually uses. Once $\rho$ supplies enough context, the 7B model recovers and then benefits from the same clutter-suppression effect that helps 3B. The marginal gains are smaller for the 7B model because its baseline is stronger to begin with.

**Win/Tie/Loss trends.** The Win/Tie/Loss breakdown in Fig. 4 clarifies this further. For Qwen2.5-VL-3B and Qwen2.5-VL-7B as the gaze-mass threshold $\rho$ increases, the win fraction is comparatively steady, the tie fraction rises, and the loss fraction falls. The rising tie indicates that enlarging the ROI primarily converts losses into ties, i.e., once the crop includes sufficient scene structure, the ROI variant often reaches the same judgment as the full-image baseline while using fewer tokens.

**Why 3B benefits more.** The absolute gain is larger for 3B. Smaller models are more susceptible to background distractors; focusing the input on gaze hotspots functions as an implicit regularizer that suppresses irrelevant context. In contrast, 7B's stronger baseline already handles clutter better but relies more on global relational cues. with very small $\rho$ it lacks layout information (high losses at 0.05–0.20), and only after crossing a context-sufficiency threshold (0.3) do losses collapse into ties. Beyond that point, both models show diminishing returns: adding periphery mostly confirms the baseline decision (more ties), while increasing tokens/FLOPs.

Table 1: Comparison of GazeVLM with ROI selection with Qwen2.5 VL baselines. FLOPs, vision tokens, and total tokens reduction are shown in parentheses.

| Model | $\rho$ | ROI-size (pixels) | Visual tokens ↓ | Total tokens ↓ | Win-rate (%) ↑ | GPT-4o score ↑ | FLOPs (G) ↓ |
|---|---|---|---|---|---|---|---|
| Qwen2.5VL-3B | – | 50.2k | 64 | 100 | – | 3.98 | 267.6 |
| GazeVLM | 0.05 | 3.01 k | 4.4 (−93.1%) | 40.4 (−59.6%) | 53.4 | 4.22 | 132.8 (−50.4%) |
| | 0.10 | 4.96 k | 6.3 (−90.2%) | 42.3 (−57.7%) | 55.7 | 4.40 | 137.6 (−48.6%) |
| | 0.20 | 8.81 k | 11.2 (−82.5%) | 47.2 (−52.8%) | 57.1 | 4.51 | 148.7 (−44.4%) |
| | 0.30 | 12.10 k | 15.6 (−75.6%) | 51.8 (−48.2%) | 61.2 | 4.45 | 158.6 (−40.7%) |
| | 0.40 | 15.28 k | 19.6 (−69.4%) | 55.6 (−44.4%) | 62.7 | 4.70 | 167.6 (−37.4%) |
| | 0.50 | 18.61 k | 23.4 (−63.4%) | 59.3 (−40.7%) | 68.2 | 4.83 | 176.1 (−34.2%) |
| | 0.60 | 21.85 k | 27.8 (−56.6%) | 63.8 (−36.2%) | 66.0 | 4.65 | 181.4 (−32.2%) |
| Qwen2.5VL-7B | – | 50.2k | 64 | 100 | - | 5.73 | 631.0 |
| GazeVLM | 0.05 | 3.01 k | 4.4 (−93.1%) | 40.4 (−59.6%) | 34.1 | 4.67 | 315.3 (−50.0%) |
| | 0.10 | 4.96 k | 6.3 (−90.2%) | 42.3 (−57.7%) | 37.7 | 5.02 | 325.3 (−48.4%) |
| | 0.20 | 8.81 k | 11.2 (−82.5%) | 47.2 (−52.8%) | 44.6 | 5.30 | 351.1 (−44.4%) |
| | 0.30 | 12.10 k | 15.6 (−75.6%) | 51.8 (−48.2%) | 50.8 | 5.84 | 374.0 (−40.7%) |
| | 0.40 | 15.28 k | 19.6 (−69.4%) | 55.6 (−44.4%) | 51.8 | 5.88 | 395.4 (−37.3%) |
| | 0.50 | 18.61 k | 23.4 (−63.4%) | 59.3 (−40.7%) | 50.9 | 5.80 | 418.2 (−33.7%) |
| | 0.60 | 21.85 k | 27.8 (−56.6%) | 63.8 (−36.2%) | 53.1 | 5.95 | 438.6 (−30.5%) |

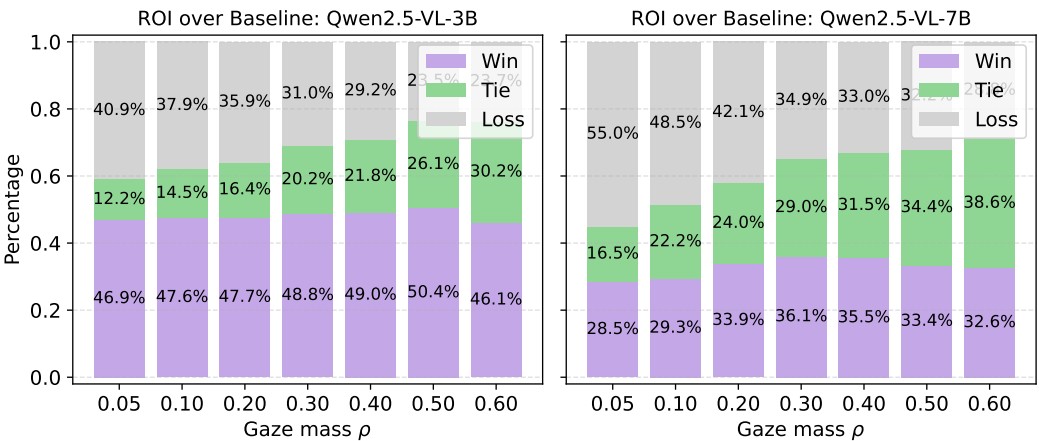

Figure 4: GPT-4o judging win/tie/loss breakdown of ROI crop input over baseline.

## 5.2 TWO-SCALE INPUT: DO WE NEED TO PRESERVE GLOBAL CONTEXT?

We augment the ROI with a low-resolution global thumbnail (28×28) to preserve coarse layout while keeping most pixels budgeted for the foveated crop. Fig. 5 compares FLOPs for ROI-only and Two-scale against the full-image baseline, and overlays the pairwise win-rate of Two-scale over ROI across gaze-mass thresholds $\rho$. Across both model sizes, Two-scale yields consistent quality gains at modest cost. The computational cost of Two-scale is slightly higher than the pure ROI because of the extra global view, yet both remain far below the dashed baseline. For Qwen2.5-VL-3B, across different crops, the win-rate stays above parity, roughly 51–57% over ROI across the sweep, indicating a net improvement without sacrificing the obvious compute advantage. For Qwen2.5-VL-7B, the pattern is similar. Two-scale adds a small, roughly overhead in FLOPs while maintaining 50-53% wins over ROI. However, the benefit is not as large as the 3B model. In short, adding a tiny global view buys quality stability while keeping a large margin to the full-image compute envelope.

The ROI delivers high-frequency detail on gaze-salient regions, and the global thumbnail supplies the low-frequency scene overview (object layout, counts, left/right relations). Many ROI-only errors

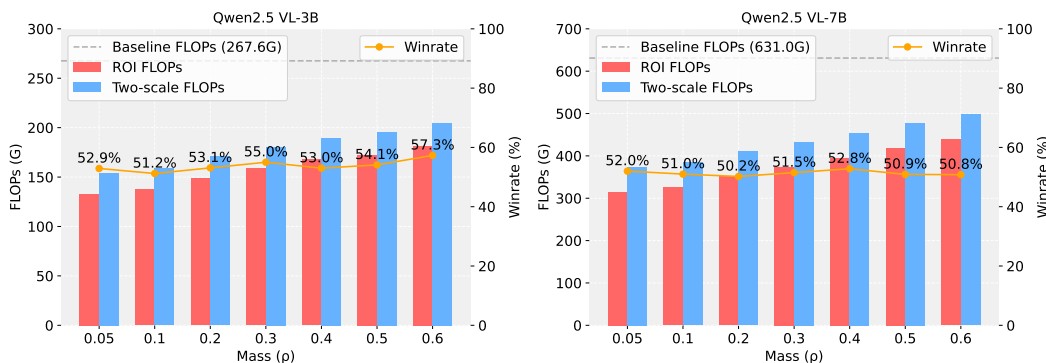

Figure 5: FLOPs (bars) and win-rate (yellow curve) of ROI-only and Two-scale inputs across gaze-mass thresholds $\rho$ on Qwen2.5-VL-3B (left) and 7B (right). Dashed line: full-image baseline.

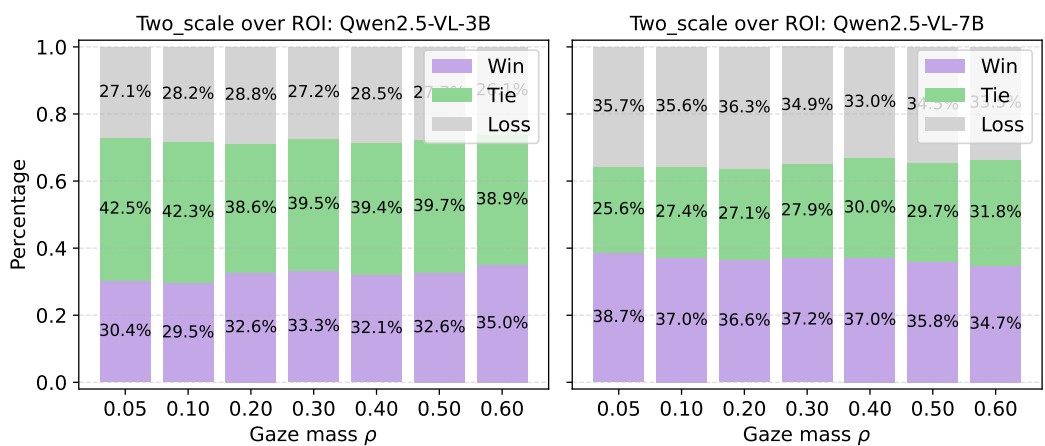

Figure 6: GPT-4o judging win/tie/loss of Two-scale input over ROI crop.

come from missing these coarse cues, the thumbnail is sufficient to convert such failures into correct or at least non-degraded decisions, which appears in the figure as steady win-rates. The gain is more pronounced for 3B: the smaller model benefits more from clutter suppression and thus from having its attention gently anchored by the global context. The 7B model, already stronger at retaining context, shows smaller but reliable improvements. It is also consistent with diminishing returns once key relations are visible.

**Win/Tie/Loss trends.** The Win/Tie/Loss breakdown in Fig. 6 clarifies this further. For both 3B and 7B models, the win fraction is essentially steady, while ties rise and losses fall. This shows that adding a tiny global thumbnail mainly converts ROI failures into ties rather than producing large win jumps, stabilizing performance at a small compute cost. The effect is stronger for 3B, which benefits more from clutter suppression and coarse layout cues, whereas 7B sees smaller but consistent gains.

## 5.3 QUALITATIVE RESULTS

We further illustrate representative qualitative examples in Fig.7. Across different cases, GazeVLM consistently focuses on gaze-relevant regions while discarding irrelevant information. Compared to the baseline full-image inference, the ROI-guided inputs lead the model to generate answers that are more precise and aligned with human intent. For instance, in fine-grained recognition tasks, the foveated crop preserves small, high-frequency details (such as the small pillow largely hidden by the sofa in case 1) that are often diluted. Meanwhile, the Two-scale design ensures that global scene layout (e.g., spatial arrangement in case 5) is maintained, preventing errors caused by missing

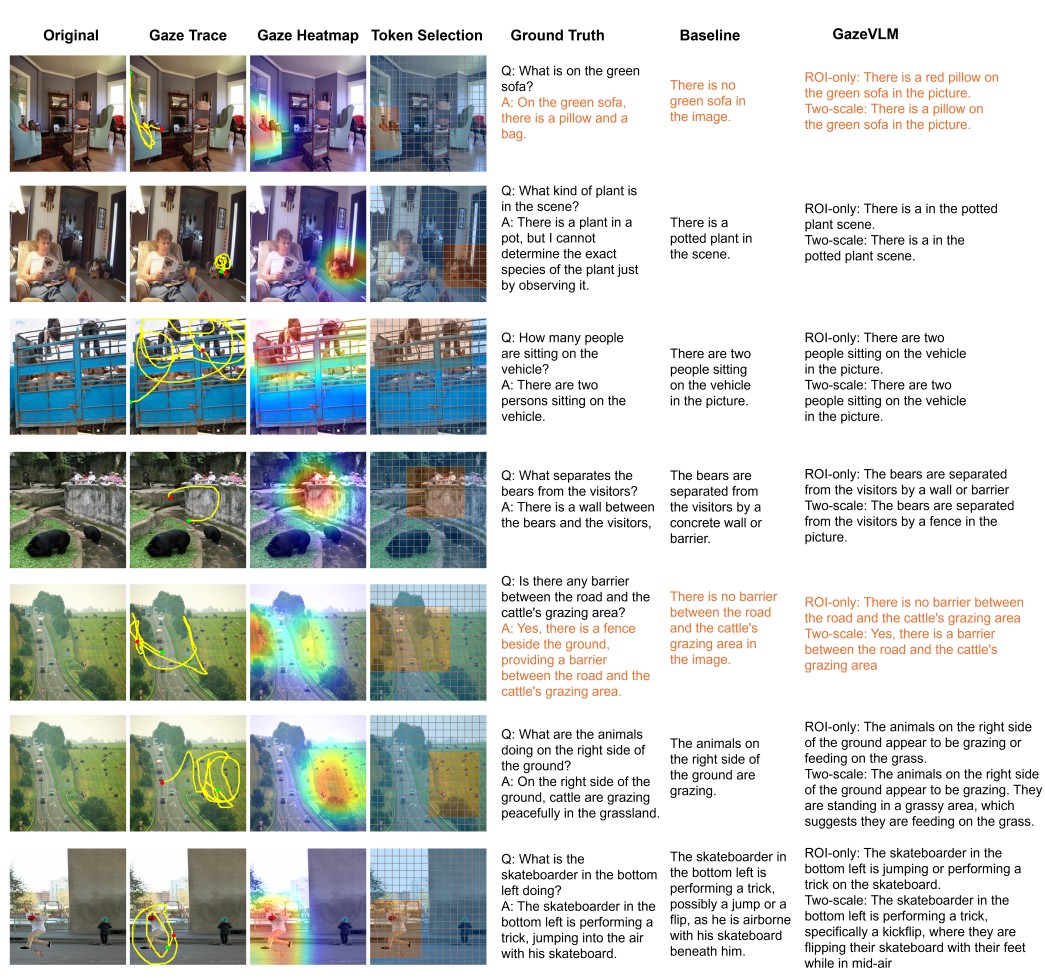

Figure 7: Qualitative results of GazeVLM (Qwen2.5-VL-3B, $\rho = 0.3$): Case 1-7. Case 1: baseline fails while both ROI-only and Two-scale succeed. Case 5: baseline and ROI-only fail, but Two-scale answers correctly. Other cases are generally ties among baseline and GazeVLM.

context. These examples highlight that GazeVLM not only reduces computation but also enhances interpretability, as the model's reasoning directly follows human attention, resulting in responses that are both efficient and semantically faithful.

# 6 CONCLUSION

In this paper, we proposed GazeVLM, a training-free and plug-and-play framework that leverages human eye gaze to guide efficient multimodal inference. By extracting gaze-driven regions of interest and combining them with a low-resolution global view, GazeVLM emulates fovea-periphery perception, enabling VLMs to allocate computation where it matters most. Experiments on VOILA-COCO with Qwen2.5-VL-3B/7B demonstrate that our approach reduces visual tokens by up to 93.1% and FLOPs by up to 50%, while maintaining or even improving answer quality relative to full-image inference. Beyond efficiency, our results highlight the promise of aligning model attention with human perceptual signals to enhance interpretability and robustness. We believe GazeVLM opens a new direction for human-in-the-loop efficient VLM design, and future work will explore its extension to streaming gaze signals, diverse modalities, and broader real-world AR/VR applications.

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
