# OpenReview forum: "Eye Gaze Tells You Where to Compute: Gaze-Driven Efficient VLMs"
_ICLR.cc/2026/Conference — Submitted to ICLR 2026_

### Official Review · Reviewer_UMDa · 2025-10-20

**Soundness:** 2
**Presentation:** 2
**Contribution:** 2
**Rating:** 2
**Confidence:** 4

**Summary:**

This paper focuses on gaze vqa in Vision-Language Models (VLMs). To address the issue of redundancy in vision tokens of VLMs, this paper presents a training-free framework that leverages the eye gaze to reduce redundant vision tokens. The proposed gaze-driven crop strategy uses gaze to crop ROIs while preserving semantic coverage. The contritbution is the proposed plug-and-play crop strategy. The authors conducted experiments on the VOILA-COCO dataset to validate the effectiveness of the proposed module.

**Strengths:**

1. The proposed method is a plug-and-play, and training-free framework.

2. The proposed strategy achieves efficiency gains with even better accuracy in visual question answering tasks.

3. This paper is well-organized and easy to read.

**Weaknesses:**

1. Insufficient evaluation. The authors only evaluated the proposed method on the Gaze VQA task. The proposed GazeVLM was only evaluated on Qwen2.5-VL-3B and Qwen2.5-VL-7B. The authors should evaluate the proposed method on more gaze-related tasks and more VLMs.

2. The crop strategy is not well studied. Although the authors have provided a series of ablation about $\rho$, there is no discussion about the other crop strategy. For example, would the external rectangle of gaze traces be useful than the proposed method?

**Questions:**

1. Is the external rectangle of the gaze trajectory more useful than the proposed method?

2. The proposed method relies on the gaze trace to crop ROI. Is it possible to crop ROI without gaze guidance? This is useful in practical applications.

---

### Official Review · Reviewer_eu2g · 2025-11-01

**Soundness:** 2
**Presentation:** 3
**Contribution:** 3
**Rating:** 2
**Confidence:** 5

**Summary:**

The authors propose GazeVLM, where human gaze is utilized to select important visual tokens serving as input to a large VLM. This approach aims to reduce latency and make VLM inference more efficient. The approach combines a low resolution global view of the image with high-definition fixated regions, much like human visual system. Hence, the method extracts the contextually relevant regions of the image that might be important for the VLM. Benchmarking on Qwen-2.5VL, the authors claim that with their method, they are able to reduce visual tokens by 93% and FLOPs by 50%, while retaining or sometimes improving VQA performance.

**Strengths:**

1. Intuitively sound idea: The core idea is novel and makes sense intuitively. The motivation behind the idea, I.e. making VLMs more efficient for edge devices is also quite pertinent in today’s world.
2. Interesting analysis: The analysis - both quantitative and qualitative - are quite well thought and meaningful.

**Weaknesses:**

1. Dataset not appropriate: As VOILA-A is based on Localized Narratives, they contain mouse traces, not actual eye gaze. While this is human attention, it is not exactly eye gaze - which is what the paper’s motivation centre around AR/VR devices hinges on. I would suggest AiR dataset [1] is a more suitable choice.

2. Experimental Section lacks transparency and detail: I have several questions regarding the experimental section: (a) Even though this paper is on improving efficiency of VLMs, I found no analysis of model throughput and inference speed. (b) The total score computation (in line 283/284) seems very arbitrary - why these weights specifically? (c) What are the averages of the individual scores of the four dimensions (coverage, accuracy, details, and fluency) generated by GPT-4o? (d) Why is win-rate not reported for the base Qwen-2.5 VL models (without GazeVLM)?

3. Absence of gaze-guided VLM baselines: The paper does not compare their method with techniques of other gaze-guided efficient VLM models, for instance  - GazeLLM [2] which crops frames in a video input based on gaze points.

[1] Shi Chen, Ming Jiang, Jinhui Yang, and Qi Zhao. Air: Attention with reasoning capability. In Computer Vision–ECCV 2020: 16th European Conference, Glasgow, UK, August 23–28, 2020, Proceedings, Part I 16, pp. 91–107. Springer, 2020.
[2] Jun Rekimoto. 2025. GazeLLM: Multimodal LLMs incorporating Human Visual Attention. In Proceedings of the Augmented Humans International Conference 2025 (AHs '25). Association for Computing Machinery, New York, NY, USA, 302–311.

**Questions:**

Please respond to my queries in weakness point 2. Additionally, I have two more questions:

1. As the text prompt might be highly correlated with the gaze, do you think with better vision-language alignment in VLMs, the text prompt will be sufficient to select relevant visual tokens?
2. What are the limitations of your approach?

---

### Official Review · Reviewer_kzN1 · 2025-11-01

**Soundness:** 3
**Presentation:** 2
**Contribution:** 2
**Rating:** 2
**Confidence:** 4

**Summary:**

The authors propose GazeVLM, a novel, training-free framework to improve the inference efficiency of Vision-Language Models (VLMs) by dynamically allocating computation based on human eye gaze. The core problem addressed is the high computational cost of VLM inference on edge devices (e.g., AR/VR headsets), which stems from redundant processing of dense, high-resolution visual tokens.

The proposed method leverages the user's eye gaze trace to identify a high-priority Region of Interest (ROI). It then constructs a "two-scale" input for the VLM, inspired by the fovea-periphery structure of human vision:

Foveal View: The high-resolution, gaze-driven ROI.

Peripheral View: A low-resolution, downsampled version of the entire global image.

This combined input is fed to an off-the-shelf VLM without any retraining. The authors evaluate their method on the VOILA-COCO benchmark using Qwen2.5-VL-3B/7B models.

**Strengths:**

- The "plug-and-play" nature of GazeVLM is a significant practical advantage. By avoiding the need for model retraining or architectural modification, this method can be applied to existing and future foundation models, drastically lowering the barrier to adoption.
- The two-scale input strategy is well-justified by its biological inspiration and is thoroughly validated in the ablation study (Section 5.2). The comparison against an ROI-only input clearly demonstrates the value of retaining global context, even at a low resolution.

**Weaknesses:**

- The entire framework is critically dependent on the availability and fidelity of the gaze signal. The paper does not discuss the method's robustness to real-world gaze-tracking imperfections, such as latency (delay between gaze and query), noise (tracker jitter), or signal loss (e.g., during blinks or fast saccades). These are non-trivial practical concerns for deployment, and the paper would be strengthened by a discussion or, ideally, a sensitivity analysis of these factors.
- The method fundamentally assumes that a user's gaze is perfectly correlated with the task-relevant visual information at the time of the query. However, gaze can be exploratory, fixated on distractors, or misaligned with the query's subject (e.g., asking "What is the weather?" while looking at a person).
- The evaluation is conducted on the VOILA-COCO dataset, which provides static gaze traces aligned with VQA pairs. While this is a reasonable starting point, it represents a "clean," offline scenario. The true value of this method is in real-time, interactive applications. It remains an open question how the system would perform with a streaming gaze signal in a dynamic environment where the user's attention and queries are fluid.
-  The experiments are confined to Visual Question Answering. While this is a strong use case, it is unclear how the method would generalize to other VLM tasks.

**Questions:**

Given that the GazeVLM framework relies on a clean, static gaze signal, could the authors please comment on its robustness to real-world imperfections such as tracker noise, system latency, and signal loss? Following this, how does the system handle "gaze-intent mismatch" scenarios, where a user's gaze is on a distractor and contradicts their verbal query? Furthermore, how do you envision adapting this method from the static, aggregated heatmaps used in the VOILA-COCO evaluation to a dynamic, real-time gaze stream in an interactive setting? Finally, could you elaborate on the generalizability of this VQA-focused approach to other key VLM tasks, such as open-ended image captioning or multi-turn dialogue, where the link between gaze and a single query is less defined?

---

### Official Review · Reviewer_fevX · 2025-11-02

**Soundness:** 3
**Presentation:** 2
**Contribution:** 2
**Rating:** 4
**Confidence:** 4

**Summary:**

This paper proposes a novel framework called GazeVLM, which aims to address the significant inference inefficiency of Vision-Language Models (VLMs) when deployed on edge devices like AR/VR glasses. The authors argue that existing VLMs suffer from substantial computational redundancy because they process all regions of an image uniformly.
The core idea of GazeVLM is to utilize the user's real-time eye gaze as a natural supervisory signal to guide the model in allocating computational resources to the user's regions of interest (ROIs). This is presented as a training-free and plug-and-play framework.
The method involves two key steps:Gaze-driven ROI Extraction:
The raw gaze trace is processed via Gaussian smoothing to generate a gaze heatmap5. A minimal bounding box that covers a specific percentage ($p$) of the total gaze mass is then extracted as the ROI6666.
Two-scale Input Representation: Inspired by the human fovea-periphery visual mechanism, GazeVLM does not just use the high-resolution ROI. Instead, it combines the ROI with a low-resolution (e.g., 28x28) global view of the entire image. The visual tokens from both views are concatenated and fed into the base VLM.
The authors conducted experiments on the VOILA-COCO benchmark using the Qwen2.5-VL-3B/7B models. Evaluation was performed using GPT-4o for pairwise judging and by measuring computational efficiency (FLOPs and token counts). The results demonstrate that GazeVLM can significantly reduce computational cost (e.g., up to 93.1% reduction in visual tokens and 50% in FLOPs) while simultaneously maintaining or even improving the answer quality compared to the full-resolution baseline.

**Strengths:**

1.Training-Free and Plug-and-Play Framework: One of the method's greatest advantages is its "plug-and-play" nature. It does not require modifying the VLM's architecture or any additional training or fine-tuning, which dramatically lowers the barrier to adoption.
2.Novel and Practical Motivation: Using human physiological signals (eye gaze) to directly guide VLM computation is a highly novel and practical idea. This is particularly relevant for the target AR/VR application scenario, as many such devices are already equipped with eye-tracking modules.

**Weaknesses:**

1.Strong Dependence on Gaze Data: This is both the core of the method and its primary limitation. The framework is entirely dependent on the availability of real-time, high-quality human gaze data. This strictly confines its application to scenarios with eye-tracking hardware (like AR/VR). The method is not applicable to the broader range of use cases where eye-tracking is absent (e.g., processing web images, video surveillance).
2.The Strong "Gaze = Intent" Assumption: The paper criticizes the "misalignment between the prompts and the region of interest" in existing methods, but GazeVLM itself relies on a strong assumption: that the user's gaze always correlates with their query's intent. However, a user could easily ask a question about a region they are not looking at (e.g., "What is behind the person I am looking at?"). In such cases, GazeVLM might erroneously crop out the critical region containing the answer.
3.Limited Methodological Novelty: While the application of gaze as a signal is novel, the technical method itself is relatively simple. The core "Two-scale" approach relies on a heuristic pre-processing step (cropping one image, downsampling another) followed by a simple token concatenation of the two resulting views. The framework does not introduce any new architectural components or learned fusion mechanisms (e.g., a dedicated adapter or cross-attention module) to explicitly integrate the global (peripheral) and ROI (foveal) information.

**Questions:**

1.Lack of Comparison to Relevant Compression Baselines: The paper's evaluation is confined to a comparison against the full-resolution baseline. To accurately situate GazeVLM's contribution, the authors must compare it against established visual token compression methods, such as SparseVLM or MMTok  mentioned in the related work. Applying these non-gaze-driven baselines to the VOILA-COCO benchmark  is necessary to provide a comprehensive analysis and isolate the specific advantages of the gaze-guided strategy over other state-of-the-art compression techniques.
2.Absence of a Critical Random Pruning Baseline: The experiments are missing a crucial control: a random visual token pruning baseline. To compellingly argue that GazeVLM's efficacy stems from its intelligent, gaze-driven selection and not merely from the act of compression itself, a direct comparison is required. This baseline should randomly discard the same number of visual tokens that GazeVLM preserves at its various $p$ thresholds (e.g., matching the 4.4 tokens at $p=0.05$ or 15.6 tokens at $p=0.3$ for the 3B model 6666). This would effectively ablate the value of the gaze signal itself.
3.Figure/Table Presentation Issues: Some figures (notably Figs. 5 and 6) use colors and scales that could be clarified, and axes/titles could be more self-contained (e.g., more explicit about x/y axes, add clearer legends to tie text and visuals together).
4.Limited Analysis of Failure Modes in Qualitative Examples: While Fig. 7 demonstrates successes, there is little discussion or illustration of failure cases, such as when GazeVLM performs worse than the baseline or when gaze misleads the model.

---

### Meta-Review · Area_Chair_PmfZ · 2026-01-06

**Summary:**

This paper introduces GazeVLM, a training-free framework that uses human eye gaze to guide Vision-Language Models (VLMs) by selectively focusing computation on gaze-driven regions of interest, combined with a low-resolution global view, to reduce visual tokens by up to 93.1% and FLOPs by 50.4% while maintaining or improving answer quality on the VOILA-COCO benchmark. Reviewers acknowledge the novel, practical motivation and plug-and-play design but highlight critical weaknesses: strong dependence on gaze data limits applicability to eye-tracking hardware; the assumption that gaze equals intent is often flawed; evaluations lack comparisons to relevant compression baselines and random pruning controls; the dataset uses mouse traces rather than real gaze; and robustness to real-world gaze noise, latency, and task generalization is unaddressed. Most reviewers gave the paper a negative review due to these methodological and experimental limitations.

**Reviewer Concerns:**

The authors did not provide a rebuttal.

**Reviewer Scores:**

The authors did not provide a rebuttal.

---

### Decision · Program_Chairs · 2026-01-26

Reject